# Freeform 3D Bioprinting Involving Ink Gelation by Cascade Reaction of Oxidase and Peroxidase: A Feasibility Study Using Hyaluronic Acid-Based Ink

**DOI:** 10.3390/biom11121908

**Published:** 2021-12-20

**Authors:** Shinji Sakai, Ryohei Harada, Takashi Kotani

**Affiliations:** Department of Materials Science and Engineering, Graduate School of Engineering Science, Osaka University, 1–3 Machikaneyama, Osaka 560-8531, Japan; harada.vn@cheng.es.osaka-u.ac.jp (R.H.); t.kotani@cheng.es.osaka-u.ac.jp (T.K.)

**Keywords:** bioprinting, freeform printing, horseradish peroxidase, choline oxidase, glucose oxidase, hyaluronic acid

## Abstract

Freeform bioprinting, realized by extruding ink-containing cells into supporting materials to provide physical support during printing, has fostered significant advances toward the fabrication of cell-laden soft hydrogel constructs with desired spatial control. For further advancement of freeform bioprinting, we aimed to propose a method in which the ink embedded in supporting materials gelate through a cytocompatible and rapid cascade reaction between oxidase and peroxidase. To demonstrate the feasibility of the proposed method, we extruded ink containing choline, horseradish peroxidase (HRP), and a hyaluronic acid derivative, cross-linkable by HRP-catalyzed reaction, into a supporting material containing choline oxidase and successfully obtained three-dimensional hyaluronic acid-based hydrogel constructs with good shape fidelity to blueprints. Cytocompatibility of the bioprinting method was confirmed by the comparable growth of mouse fibroblast cells, released from the printed hydrogels through degradation on cell culture dishes, with those not exposed to the printing process, and considering more than 85% viability of the enclosed cells during 10 days of culture. Owing to the presence of derivatives of the various biocompatible polymers that are cross-linkable through HRP-mediated cross-linking, our results demonstrate that the novel 3D bioprinting method has great potential in tissue engineering applications.

## 1. Introduction

Three-dimensional (3D) printing, also known as additive manufacturing or rapid prototyping, has been driving innovations in various fields, such as manufacturing, education, and medicine. Bioprinting is a subcategory of 3D printing and is defined as the technology for the fabrication of biological tissues and organs. It allows the fabrication of cell-laden constructs with spatially complex structures that are otherwise difficult to fabricate using conventional biofabrication techniques [1,2]. In bioprinting, the ink that forms cell-laden hydrogels in the manufacturing process has gained much popularity [3]. A hydrogel is composed of a network of cross-linked hydrophilic polymer chains in an aqueous solution. Various solutions of polymers, such as alginate [4,5,6], hyaluronic acid [7,8], gelatin [6,9,10], chitosan [11,12], silk fibroin [13,14], and collagen [15], including their derivatives, have been used as the ink. 

An advantage of the use of ink in hydrogel constructs is the possibility of controlling the microenvironment surrounding the cells [3]. While the required properties may be different for each type of cell, favorable cellular functions, such as proliferation, differentiation, migration, and organization, can be enhanced by tuning the biochemical and mechanical cues of hydrogels [3,16,17,18,19]. Additionally, the biodegradation rate and tissue response in vivo can also be controlled by tuning the degree of cross-linking and polymer composition [3].

A notable feature of the ink used in the fabrication of cell-laden hydrogels is that the resulting constructs are softer and deform more easily by their weight when formed in the air than the non-hydrogel constructs used in industrial 3D printing. Deformation can occur even during the manufacturing process and can be an obstacle to the fabrication of large constructs with good shape fidelity to blueprints. A possible way to reduce the chances of deformation is to use ink that forms harder hydrogels by increasing the cross-linking density and polymer content [18]. However, the effect of increased hydrogel stiffness on cellular behavior would need further consideration [3,19,20,21]; cell spreading and proliferation may be restricted due to increased matrix stiffness [19]. 

Freeform bioprinting, in which cell-suspending ink is extruded into a support bath composed of supporting materials, has attracted much attention as a promising technique for fabricating cell-laden soft hydrogel constructs [22,23]. The supporting materials function to provide physical support during the 3D printing process, and the gelation of the embedded ink progresses in the support bath. Therefore, freeform bioprinting enables the preparation of soft constructs with good shape fidelity compared to conventional printing in the air [22,24]. To date, various gelation techniques have been applied for the gelation of ink in freeform bioprinting, namely ionic gelation of alginate [22], pH change-induced gelation of collagen [22], photo-cross-linking of methacrylated polymers [25], and enzymatic gelation of fibrinogen by thrombin [22]. We believe that the development of a novel freeform bioprinting process involving cytocompatible gelation, in which various inks may be applicable, would further enhance the usefulness of the method. 

This study aimed to report a freeform bioprinting process involving gelation mediated by the enzymatic cascade reaction of choline oxidase (COD) and horseradish peroxidase (HRP) (Figure 1). COD catalyzes the conversion of choline and oxygen into betaine glycine and hydrogen peroxide. HRP further catalyzes the formation of covalent bonds between phenolic hydroxyl (Ph) groups through oxidation in presence of H_2_O_2_. HRP-mediated gelation has been extensively studied as a promising method for fabricating cell-laden hydrogels for biomedical applications [26]. Various derivatives of polymers, including gelatin [27], chitosan [28], and hyaluronic acid [29] have been applied to gelation, and hydrogel property-dependent cellular behavior has been reported [30,31,32,33]. The most common way to supply H_2_O_2_ for HRP-mediated gelation is to add an aqueous H_2_O_2_ solution to the mix containing HRP and polymer(s) possessing Ph groups (polymer-Ph) [26]. Owing to the dose- and time-dependent cytotoxicity of H_2_O_2_ [34,35,36,37], the amount of H_2_O_2_ supplied to obtain cell-laden hydrogels should be carefully determined, as reported previously [30,31,32,33]. Intuitively, as a bioprinting system that combines freeform bioprinting with HRP-mediated gelation, it would seem feasible to extrude an ink containing HRP or H_2_O_2_ and polymer-Ph into a support bath containing H_2_O_2_ or HRP, respectively. However, the cytotoxic effect of H_2_O_2_ on cells increases with increasing printing time in these systems and would possibly pose a critical challenge while printing larger tissues and organs for practical applications. This is due to the longer exposure time of cells to H_2_O_2_ in the support bath during printing or in the ink before extrusion. 

The bioprinting process proposed here involves the extrusion of a bioink composed of polymer-Ph, cells, HRP, and choline into a support bath containing COD. The potential adverse effects of H_2_O_2_ exposure on cells can be suppressed by controlling the content of choline in the ink, instead of using ink containing COD and a support bath containing choline. Choline is an essential nutrient that regulates body function and human health. Glucose oxidase (GOD) and HRP are enzyme pairs used to obtain hydrogels from aqueous polymer-Ph solutions for biomedical applications [38,39,40,41]. We believe that the use of COD, instead of GOD, would decrease the risk of cell damage due to H_2_O_2_ produced by enzymes remaining in the printed constructs when the printed cell-laden constructs are incubated in a medium. The choline content in conventional media, such as Dulbecco’s modified Eagle medium (DMEM), minimum essential medium, and RPMI 1640, is less than 1/100 of glucose. 

The objective of the current study was to demonstrate the remarkable potential of the freeform bioprinting process, involving the COD/HRP-mediated gelation of ink. A solution containing a derivative of hyaluronic acid possessing Ph groups (HA-Ph) and mouse fibroblasts was used as a model ink to investigate the feasibility of the printing method. Xanthan gum (XG) was used as supporting material in the support bath for embedding the ink. The usefulness of XG was recently reported by Patrício et al., who fabricated ionically cross-linked cell-laden Ca-alginate hydrogel constructs by extruding a cell-suspending sodium alginate solution in an XG support bath containing calcium ions [42]. Establishing a method that enhances the freeform bioprinting process will foster advances toward the fabrication of cell-laden soft hydrogel constructs in the desired shape.

## 2. Materials and Methods

### 2.1. Materials

Choline chloride, glucose, N-hydroxysuccinimide (NHS), COD from *Alcaligenes* sp. (17 U mg^−1^), hyaluronidase from bovine testes, and HRP (200 U mg^−1^) were purchased from Fujifilm Wako Pure Chemical (Osaka, Japan). Water-soluble carbodiimide hydrochloride (WSC) and GOD from *Aspergillus niger* (360 U mg^−1^) were purchased from Peptide Institute (Osaka, Japan) and SERVA Electrophoresis GmbH (Heidelberg, Germany), respectively. Xanthan gum (1.45–2.0 Pa s at 1% in water, 20 °C), sodium hyaluronic acid (molecular weight of circa 1000 kDa), and tyramine hydrochloride were obtained from Tokyo Chemical Industry (Tokyo, Japan), Kewpie (Tokyo, Japan), and Chem-Impex International (Wood Dale, IL, USA), respectively. HA-Ph was synthesized by conjugating hyaluronic acid and tyramine using WSC and NHS based on a previously reported method [39]. The content of Ph groups in HA-Ph was 5–7 × 10^−5^ mol-Ph/g. Fibroblast derived from a mouse embryo, 10T1/2 cells, were obtained from the Riken (Ibaraki, Japan). The cells were cultured in DMEM containing 10% (*v*/*v*) fetal bovine serum (FBS) at 37 °C in humidified air containing 5% CO_2_.

### 2.2. Effect of COD and GOD in Cell Medium 

Cells were seeded in a 12-well cell culture dish at a density of 2.1 × 10^3^ cells/cm^2^. After overnight culture, the medium was replaced with a medium containing COD at 0.1–2.5 U/mL. At 24 h after the COD addition, the medium was replaced with a fresh one not containing COD, and the cells were cultured for an additional 48 h. The growth profile of the cells was determined using an incubation monitoring system (CM20; Olympus, Tokyo, Japan). As controls, growth profiles of cells in media with GOD and without COD and GOD were determined. 

### 2.3. Gelation Time

The time necessary for gelation of the HA-Ph solution was measured in a phosphate-buffered saline (PBS, pH 7.4) containing 1.25% (*w*/*v*) HA-Ph, based on a previously reported method [43,44]. Briefly, a solution containing HA-Ph, COD, and HRP (262.5 µL) was poured into the wells of a 48-well plate and stirred using a magnetic bar. Then, PBS containing choline chloride (37.5 µL) was added to the wells. The gelation was signaled when magnetic stirring was hindered and the surface of the solution swelled. The concentrations of COD, HRP, and choline chloride were varied in the range of 0.05–3.0 U/mL, 0.1–10 U/mL, and 0.1–50 mM, respectively.

### 2.4. Preparation of Support Bath

COD was dissolved in phosphate-buffered saline (PBS). XG 1.0% (*w*/*v*) was then slowly added to the solution with vigorous stirring at 25 °C. The resultant slurry was poured into a plastic container of 40-mm diameter and 40-mm height and in a cell culture dish of 60-mm diameter. These containers were used as support baths for the printing of cell-free and cell-laden constructs, respectively. The COD content in the support baths for printing cell-free constructs was 0, 0.1, or 0.5 U/mL, and that for printing cell-laden constructs were 0.5 U/mL. 

### 2.5. Bioprinting

Three-dimensional hydrogel constructs were printed using an extrusion-based 3D printing system developed by modifying a commercial 3D printer (Anycubic i3 Mega; Anycubic, Guangdong, China). Ink containing HA-Ph, HRP, and choline chloride at 1.25% (*w*/*v*), 0.1 or 5 U/mL, and 5 mM, respectively, with a viscoelastic property shown in Appendix A, was used for printing cell-free constructs. The content of HRP in the ink used in studies with cells was 5 U/mL. For studies using cells, 10T1/2 cells were suspended in the ink at 5.0 × 10^5^ cells/mL. The ink was extruded from a 27-gauge stainless steel needle at 34 mm/s with 0.3 mm height of each layer, into the support bath. The support bath was fixed on a platform and moved at 17 mm/s at 25 °C. 

In the studies using cells, cell-suspending ink was extruded in the support bath to obtain disc-shaped cell-laden hydrogel constructs (8-mm diameter, 0.3-mm height). After 5 min of standing, the resultant constructs were washed with PBS to remove XG and incubated in the medium at 37 °C. After 1 d of incubation, the cell-laden hydrogel constructs were soaked in a medium containing 2 mg/mL hyaluronidase in a 6-well cell culture dish for degradation. Growth profiles of cells released from the degraded hydrogel constructs were determined by measuring the area occupied by adherent cells using commercial software (CKX-CCSW, Olympus, Tokyo, Japan). Cells not exposed to the printing process were used as controls. In addition to the studies on cells released from the degraded hydrogel constructs, changes in morphology and viability of cells enclosed in the printed hydrogel constructs for 10 days were observed using a fluorescence microscope (BZ-9000, Keyence, Osaka, Japan) by staining with calcein-AM (Dojindo, Kumamoto, Japan) and propidium iodide (PI) (Dojindo, Kumamoto, Japan). 

## 3. Results and Discussion

### 3.1. Effect of COD and GOD in Cell Medium 

GOD was the first candidate to provide H_2_O_2_ for HRP-mediated cross-linking since the usefulness of GOD/HRP-mediated reactions for the fabrication of hydrogels in biomedical applications had been previously reported in both in vitro and in vivo conditions [38,39,40,41]. In the bioprinting of cell-laden constructs using inks containing enzymes, complete removal of the enzymes from the resultant constructs, without excessive time-consuming rinsing, was difficult. Therefore, the potential adverse effects of the remaining enzymes needed to be considered. We first examined the effects of GOD and COD in the cell medium by continuously culturing the cells in a medium containing GOD or COD for 24 h and in a normal medium for an additional 48 h.

As shown in Figure 2a,d, the cells cultured in the medium containing GOD did not grow during the two sequential periods, regardless of the GOD content. In contrast, the cells cultured in a medium containing COD grew similar to those cultured in a medium free of the enzyme during the first 24 h. Furthermore, there was no noticeable growth inhibition and no significant difference in morphology in the subsequent 48 h of incubation (Figure 2b,c,e). 

Growth inhibition in the presence of GOD could be explained by the H_2_O_2_ generated by GOD from 5.6 mM glucose contained in DMEM; the choline content in DMEM was 0.03 mM. Theoretically, the same amounts of H_2_O_2_, as glucose and choline in the medium, are generated by catalytic oxidation mediated by GOD and COD, respectively. Due to the antioxidant activity of albumin in FBS [45], the exact amount of H_2_O_2_ that affected cells was unclear. However, the current result seemed consistent with that reported by Park for human fibroblasts [37], where the growth rate of cells exposed to >0.1 mM H_2_O_2_ for 24 h was approximately 20%, while that of cells exposed to <0.05 mM H_2_O_2_ for 24 h was almost the same as that of non-exposed cells. Based on these results, we decided to use COD as a source of H_2_O_2_ for HRP-mediated cross-linking in the studies presented below.

### 3.2. Gelation Time

In freeform bioprinting, ink is cured once it is extruded into the support bath from an extruder needle. Rapid gelation of the ink helps to stabilize the printed structure, preventing unwanted deformations and diffusion during the printing process [24]. Therefore, the effects of factors on gelation time should be determined. We studied the effects of COD, HRP, and choline content on the time required for gelation. 

As shown in Figure 3, gelation time decreased with increasing COD content from 0.05 U/mL (110 s) to 0.5 U/mL (9 s) at 5 U/mL HRP and 5 mM choline (Figure 3a). A further increase in COD content to 1 and 3 U/mL did not decrease the gelation time additionally. The increase in HRP from 0.1 to 5 U/mL and choline from 0.01 to 5 mM also decreased gelation time (Figure 3b,c); however, further doubling to 10 U/mL and 10 mM, respectively, did not decrease the values additionally (*p* = 0.63 and 0.79, respectively).

The trends indicating a decrease in gelation time with increasing COD, HRP, and choline contents, observed at certain content ranges, are consistent with those reported for the solution of alginate derivatives possessing Ph groups [46]. The trends can be explained by the general theory of enzyme kinetics with no inhibition. The absence of a decrease in gelation time when the COD, HRP, and choline contents were beyond 0.5, 5, and 5 mM, respectively, could be explained by the time required for the diffusion of choline that was added to initiate the reaction. 

### 3.3. 3D Printing 

The possibility of printing 3D hydrogel constructs using the method described here was investigated by printing cylindrical hyaluronic acid hydrogel constructs with a 15-mm diameter, 1.5-mm wall thickness, and 10-mm height (Figure 4a) under several conditions. As shown in Appendix A, the ink, with 1.25% (*w*/*v*) HA-Ph, 5 U/mL HRP, and 5 mM choline, embedded in a support bath without COD, did not maintain its shape when the supporting material was diluted with PBS. In contrast, a cylindrical hydrogel construct was retained after washing with PBS when the same ink was embedded in a support bath containing 0.5 U/mL COD (Figure 4b–e). The shape of the resultant cylindrical construct was almost identical to that of the blueprint. The resultant hydrogel construct maintained its shape in PBS for more than 1 week (Figure 4f). Interestingly, the hydrogel construct had a layered structure consisting of individual hydrogel threads (within the white solid box in Figure 4f). Cylindrical constructs were obtained even under conditions where either the COD content in the support bath (Appendix A) or the HRP content (Appendix A) in ink was reduced to 0.1 U/mL. However, the layered structure was found only in the construct obtained using the support bath with 0.1 U/mL COD (Appendix A). The construct obtained using the ink with 0.1 U/mL HRP was weak and deformed during soaking in PBS (Appendix A). We also fabricated a conical hyaluronic acid hydrogel construct with 20-mm diameter, 2-mm wall thickness, and 10-mm height (Figure 4g); as shown in Figure 4h–l, hence a hydrogel construct with good shape fidelity to the blueprint, stable for at least 9 days in PBS, and with the visible layered structure of individual hydrogel threads was obtained. Freeform printing enabled the fabrication of a conical construct with the apex facing downward. Without using supporting materials, soft hydrogel constructs cannot be fabricated in the air using conventional extrusion 3D printing. The different appearance of the conical constructs before (Figure 4h,i) and after washing with PBS (Figure 4j,k) is due to the removal of the supporting material, which caused the cylindrical constructs to tilt. 

The results demonstrated that 3D hydrogel constructs can be printed via a freeform bioprinting process involving COD/HRP-mediated ink gelation. The importance of setting the COD content in the support bath and HRP content in the ink was also demonstrated. The condition-dependent appearance of the layered structure of individual hydrogel threads and the ease of deformation, indicating the softness of the construct, can be explained by the correlation with the time duration for gelation of the embedded ink. It has been reported that the viscoelastic properties of inks govern the printability and shape fidelity of bioinks in bioprinting [47,48]. Since the viscoelastic properties of the inks used in this study were similar, regardless of the contents of HRP and choline (Appendix A), it is not necessary to consider the effect of the viscoelastic properties. Slow gelation of ink causes the adjacent ink threads to fuse since the extruded ink diffuses into the surrounding slurry until gelation. Gelation of diffused ink, which lowers the density of the cross-linked polymer, induces the formation of a soft hydrogel. As shown in Figure 3, the gelation time detected for 1.25% (*w*/*v*) HA-Ph + 5 mM choline chloride solution containing 0.1 U/mL HRP + 0.5 U/mL COD (105 s) was more than ten times and twice as long than those detected for the solutions containing 5 U/mL HRP + 0.5 U/mL COD (9 s) and 5 U/mL + 0.1 U/mL COD (42 s), respectively. Although measurement of gelation time of the extruded ink in a support bath was difficult, Figure 3 could help us to qualitatively understand the magnitude of the relationship of the extruded ink with the condition-dependent gelation time. Gelation of ink containing 1.25% (*w*/*v*) HA-Ph, 5 mM choline chloride, and 5 U/mL HRP extruded in a support bath containing 0.5 U/mL COD should be faster than that extruded in the support bath containing 0.1 U/mL COD, and can result in a hydrogel with a higher density of cross-linked HA-Ph. In the following experiments, ink containing 1.25% (*w*/*v*) HA-Ph, 5 mM choline chloride, 5 U/mL HRP, and a support bath containing 0.5 U/mL COD were used. 

We also studied the effect of the 3D printing process, involving extrusion from a microneedle and COD/HRP-mediated ink gelation in a support bath, on cells. The cells released from the printed HA-Ph hydrogels, through degradation by soaking in a medium containing hyaluronidase, were round in shape and floated in the medium after 20 min of treatment (Figure 5a). After 100 min of treatment, we found the cells attached to the cell culture dish (Figure 5b). The cells looked elongated, similar to those not exposed to the printing process (Figure 2c and Figure 5c). Cells released from the degraded constructs showed the same growth profile as those that were not exposed to the printing process (Figure 5d). Additionally, the fact that cells were released from the printed HA-Ph hydrogel constructs by treatment with hyaluronidase, indicated that the HA-Ph hydrogels obtained through the printing method were biodegradable, similar to those obtained through direct H_2_O_2_-addition [29] and GOD/HRP-mediated gelation [39] processes.

Finally, we investigated the behavior of cells in printed hydrogel constructs. As shown in Figure 6a–d, the enclosed cells showed a round shape during 10 days of culture, with no obvious growth. The viability of the cells immediately after printing was 98% (Day 0; Figure 6e), and after 10 days of culture was 94%. The results shown in Figure 5a–c and the viability immediately after printing, indicated that the exposure of mouse fibroblast 10T1/2 cells to COD/HRP-mediated ink gelation and embedding of the ink in the support bath did not have adverse effects leading to immediate cell death. Furthermore, comparable growth of the cells released from the degraded constructs with those not exposed to the printing process (Figure 5d), in addition to the more than 85% viability of the enclosed cells during 10 days of culture (Figure 6e), indicated that the bioprinting method described here does not show adverse effects resulting in delayed cell death. The overall results confirmed that the bioprinting method was cytocompatible. 

Regarding the round shape and non-obvious growth of cells in the printed HA-Ph hydrogel constructs (Figure 6a–d), the results were not specific to the method described here. A similar result was reported for the 10T1/2 cells enclosed in HA-Ph hydrogel constructs fabricated through HRP-mediated gelation in air containing H_2_O_2_ [8]. The non-cell adhesive nature of hyaluronic acid-based hydrogels is well known [49,50]; the properties required for cell-enclosing hydrogels, especially cell adhesiveness, are different for each cell type and application. Although it is beyond the scope of this study, the use of gelatin derivatives possessing Ph groups [27,51], instead of HA-Ph or with HA-Ph [52], could be a possible approach for inducing both elongation and growth of enclosed adherent cells. The mechanical properties of the hydrogel surrounding the cells also need to be set appropriately, as it has been reported to affect cell functions, morphology, and growth [19,53]. An attractive feature of COD/HRP-mediated gelation and freeform bioprinting is the abundance of polymer-Ph groups derived from a variety of polymers [26] which can be potentially applied to this method. Silk fibroin contains approximately 5% tyrosine residues possessing Ph groups [54] and is also a candidate material for the ink. Further investigation of the use of these materials will be conducted in the future to examine their specific applications. 

## 4. Conclusions

We proposed a new 3D bioprinting method, based on the gelation of bioink mediated by COD, HRP, and freeform printing. In this method, choline, HRP, and polymer-Ph were included in the ink. COD was placed in the support bath. We demonstrated that the risk of damaging cells due to the unwanted generation of H_2_O_2_ by the oxidase remaining in the printed hydrogel constructs can be reduced by using COD, but not GOD. We also demonstrated that the method enabled printing of HA-Ph hydrogel constructs with a geometry impossible to fabricate otherwise by conventional 3D printing extrusion of soft hydrogel constructs in air. Additionally, we confirmed cytocompatibility of the bioprinting method based on the results indicating comparable growth of mouse fibroblast cells, released from the printed hydrogels through treatment with hyaluronidase on cell culture dishes, with those not exposed to the printing process, and more than 85% viability of the enclosed cells during 10 days of culture. These results demonstrated that the method proposed in the present study could serve as a promising technique. Thus, the novel 3D bioprinting method developed herein has marked potential for use in tissue engineering.

## Figures and Tables

**Figure 1 biomolecules-11-01908-f001:**
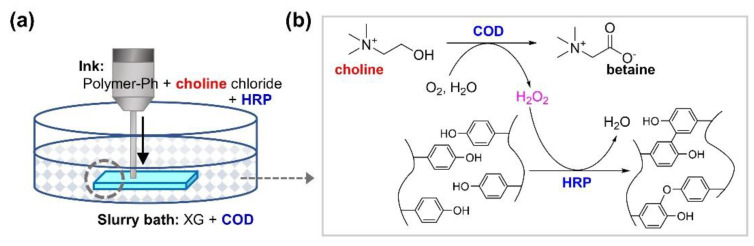
(**a**) Schematic drawing of choline oxidase (COD) and horseradish peroxidase (HRP)-mediated freeform 3D printing in a support bath containing xanthan gum (XG) and COD by embedding ink containing polymer possessing phenolic hydroxyl groups (Polymer-Ph), choline chloride, and HRP. (**b**) Chemical reaction scheme of cross-linking of phenolic hydroxyl groups through HRP-catalysis, consuming H_2_O_2_ generated through COD-catalyzed oxidation of choline.

**Figure 2 biomolecules-11-01908-f002:**
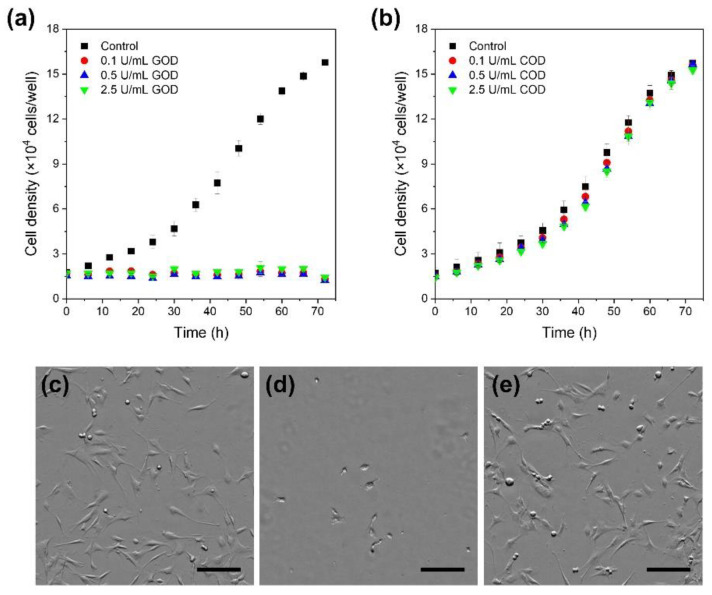
Growth profiles of murine fibroblast 10T1/2 cells: (**a**) Glucose oxidase (GOD) or (**b**) choline oxidase (COD) was included in the medium during the first 24 h at concentrations shown in each panel. Control: cultured in a medium free of GOD and COD. Morphology of 10T1/2 cells at 40 h: (**c**) control, (**d**) with 2.5 U/mL GOD as shown in panel (**a**), and (**e**) with 2.5 U/mL COD as shown in panel (**b**). Bars in panels (**a**,**b**): standard deviation (*n* = 3), and in panels (**c**–**e**): 200 µm.

**Figure 3 biomolecules-11-01908-f003:**
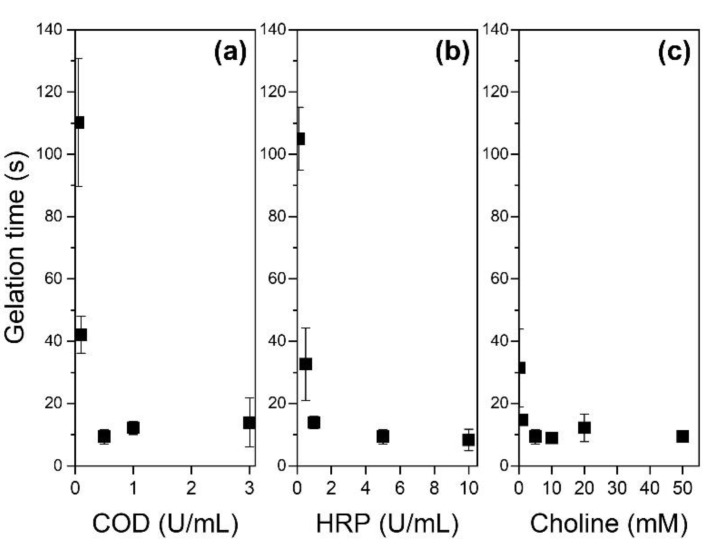
Effects of (**a**) choline oxidase (COD), (**b**) horseradish peroxidase (HRP), and (**c**) choline concentrations on gelation time of 1.25% (*w*/*w*) HA-Ph solution. The HRP content in the studies shown in panels (**a**,**c**) was 5 U/mL, the COD content in the studies shown in panels (**b**,**c**) was 0.5 U/mL, and the choline content in the studies shown in panels (**a**,**b**) was 5 mM. Bars: standard deviation (*n* = 4).

**Figure 4 biomolecules-11-01908-f004:**
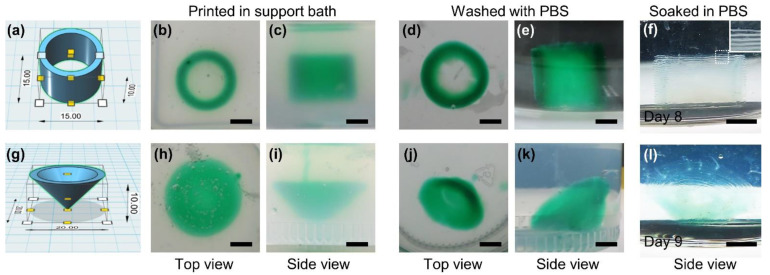
Blueprints of (**a**) cylindrical and (**g**) conical 3D constructs, and (**b**,**d**,**h**,**j**) top and (**c**,**e**,**f**,**i**,**k**,**l**) side views of HA-Ph hydrogel constructs obtained by extruding an ink containing 1.25% (*w*/*v*) HA-Ph, 5 U/mL HRP, and 5 mM choline chloride in PBS into a support bath containing 1.0% (*w*/*v*) XG and 0.5 U/mL COD in PBS. The constructs (**b**,**c**,**h**,**i**) just after printing in supported bath, (**d**,**e**,**j**,**k**) after washing with PBS for removing XG, and after (**f**) 8 and (**l**) 9 days of soaking in PBS. The image with a solid white border in panel (**f**) is a magnified view of the part with a dashed border in the same panel. Bars: 5 mm.

**Figure 5 biomolecules-11-01908-f005:**
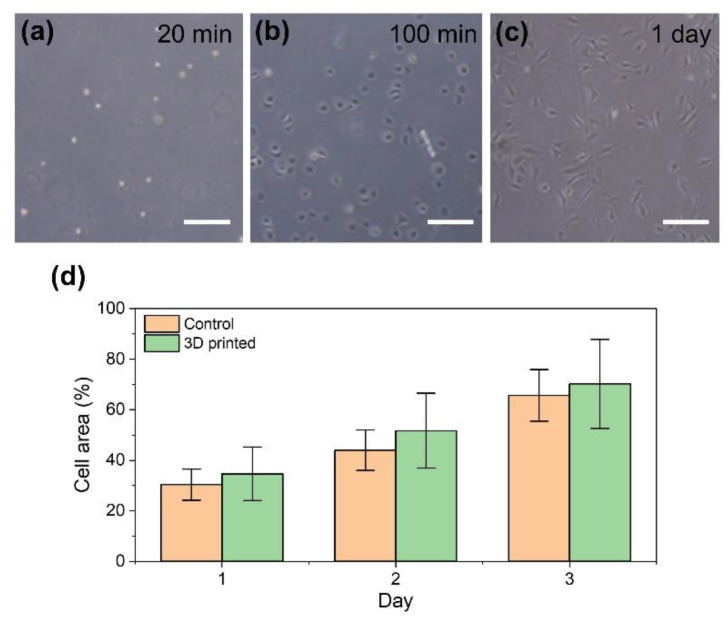
Mouse fibroblast 10T1/2 cells released on a cell culture dish from degraded cell-laden HA-Ph hydrogel by soaking in a medium containing hyaluronidase. Cell morphologies at (**a**) 20 min, (**b**) 100 min, and (**c**) 1 day after treatment for degradation. Growth profiles of the released cells (3D printed) and of those not exposed to 3D printing (control) are expressed in terms of cell area on the cell culture dish. Bars in panels (**a**–**c**): 200 µm and panel (**d**): standard deviation (*n* = 3).

**Figure 6 biomolecules-11-01908-f006:**
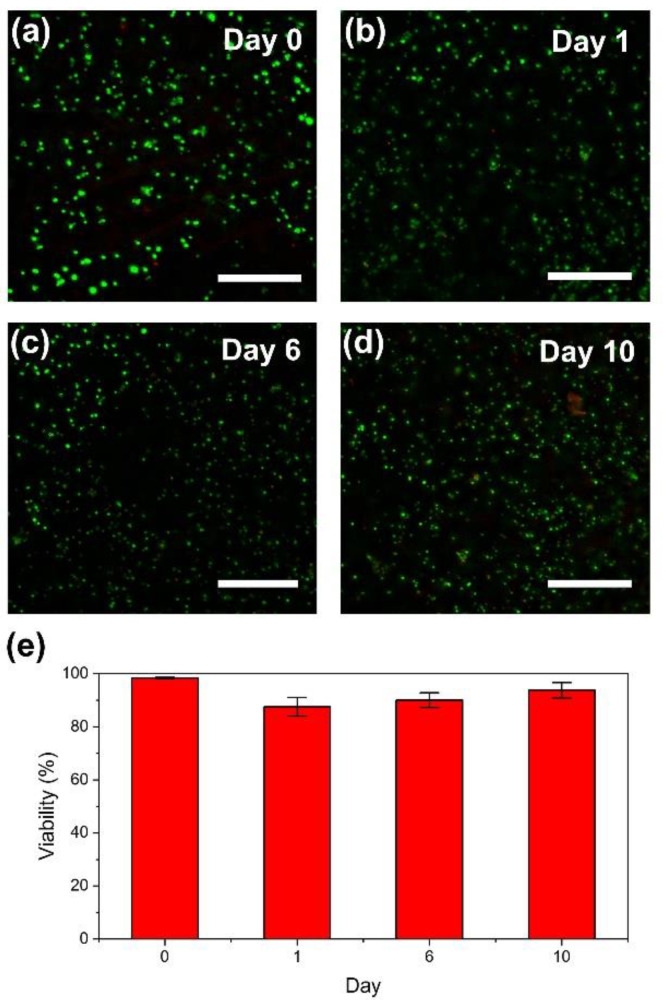
(**a**–**d**) Fluorescence microphotographs of mouse fibroblast 10T1/2 cells enclosed in bioprinted HA-Ph hydrogel discs: (**a**) immediately after printing, at (**b**) 1 day, (**c**) 6 days, and (**d**) 10 days after printing. Live and dead cells showed green fluorescence, attributed to Calcein-AM, and red fluorescence, attributed to PI, respectively. (**e**) Viabilities of the enclosed cells. Bars in panels (**a**–**d**): 200 µm, and in panel (**e**): standard deviation (*n* = 3).

## Data Availability

All experimental data within the article and its Appendix A are available from the corresponding author upon reasonable request.

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
