# Peer review of "Freeform 3D Bioprinting Involving Ink Gelation by Cascade Reaction of Oxidase and Peroxidase: A Feasibility Study Using Hyaluronic Acid-Based Ink"

_biomolecules, 2021, doi:10.3390/biom11121908_

Round 1

Reviewer 1 Report

This paper designs a novel method of 3D printing by using COD/HRP-mediated gelation of ink, which cleverly solves the problem of cytotoxicity produced by hydrogen peroxide. A confocal microscopic study of the cells in the hydrogel cultured for several days is suggested for demonstration of the success cell-laden printing.  Or the cell status before degradation by hyaluronidase should be provided by flow cytometry. The results as shown in figure 5 and 6 alone seemed not enough.  Furthermore, the conclusion section should be concisely rewritten. The main findings should be emphasized without so many discussions.

Author Response

Response to Reviewer 1 Comments

Comment 1. A confocal microscopic study of the cells in the hydrogel cultured for several days is suggested for demonstration of the success cell-laden printing.  Or the cell status before degradation by hyaluronidase should be provided by flow cytometry. The results as shown in figure 5 and 6 alone seemed not enough.

Response 1. Thank you for the comment. As pointed by the reviewer, we did not use a confocal microscope but used a fluorescence microscope (BZ-9000, Keyence, Osaka)(https://www.americanlaboratory.com/617-News/19006-BZ-9000-All-in-one-Fluorescence-Microscope/), for observing the cells in the printed constructs for 10 days. As mentioned in the above web site, and we had also confirmed by ourselves, the viability data obtained using the fluorescence microscope is comparable to the data obtained by the confocal microscope of our laboratory.     

Comment 2. The conclusion section should be concisely rewritten. The main findings should be emphasized without so many discussions.

Response 2. Thank you for the very valuable suggestion. We have revised the conclusion section as suggested by the reviewer. We have deleted discussions and emphasized the main findings.

Reviewer 2 Report

Although the topic of the manuscript is of interest and relevance for the field of bioprinting, the authors should address some points.

  • The quality of the scientific writing should be improved
  • The methods section is not detailed
  • Please use cells/cm2 rather than cells/well
  • Biochemical and biophysical properties of the ink/hydrogel should have been assessed (e.g mechanical properties, rheological properties, cross-linking ratio)
  • Figure 2 legend, “murine” instead of marine
  • It would be interesting to see data obtained with human cells, in addition to murine cells. This would increase the relevance of this study
  • I would like to read results and discussion in 2 separate paragraphs. The conclusions are too long.
  • The mechanical properties of the material could affect cell morphology/growth. This could be an element for the discussion

Author Response

Response to Reviewer 2 Comments

Comment 1. The methods section is not detailed.

Response 1. We have checked the methods section and have added details (Line 132-133; 138-40; 158-159; 164).

Comment 2. Please use cells/cm2 rather than cells/well

Response 2. In accordance with the comment, we have revised to be cells/cm2 (Line 137).

Comment 3. Biochemical and biophysical properties of the ink/hydrogel should have been assessed (e.g mechanical properties, rheological properties, cross-linking ratio)

Response 3. Thank you for the suggestion. In accordance with this comment, we have measured a viscoelastic property of ink and have added the data as Figure S1 with the relating description in Materials and method section (Line 174-175) and Result and Discussion section (Line 298-302). The information of the instrument used for measuring the property is shown in the legend of Figure S1. Regarding the properties of the printed hydrogel constructs. We attempted to measure Young's modulus by compressing the construct but could not get reliable data because of delamination between the laminated layers shown in Figure 4f and 4l. We think the probe smaller than individual hydrogel filament (<0.2 mm diameter) is necessary for compressing individual printed hydrogel filament. Unfortunately, at present, we do not find and have the instrument for the measurement. Thus, we can not add the data in this manuscript.

Comment 4. Figure 2 legend, “murine” instead of marine

Response 4. Thank you for the indication. We have revised it.

Comment 5. It would be interesting to see data obtained with human cells, in addition to murine cells. This would increase the relevance of this study

Response 5. Thank you for the good suggestion. We completely agree with it. This study aims to show the feasibility of the proposed method using mammalian cells. We will do the study of using human cells in future.

Comment 6. I would like to read results and discussion in 2 separate paragraphs.

Response 6. Biomolecules has papers that include a results & discussion section. We also believe that our results can be better understood with a results & discussion section.

Comment 7. The conclusions are too long.

Response 7. In accordance with this comment and Reviewer 1’s comment 2, we have revised the conclusion section. We have deleted discussions and emphasized the main findings.

Comment 8. The mechanical properties of the material could affect cell morphology/growth. This could be an element for the discussion

Response 8. Thank you for the suggestion. We have added the description about the effect of mechanical properties on cells in text (Line 358-359).

Reviewer 3 Report

This manuscript by Shinji et.al. mainly described a gelation method by cascade reaction of oxidase and peroxidase process. They used this method to cross choline horseradish peroxidase (HRP) and hyaluronic acid derivative as the bioink to do the freeform bioprinting in the Xanthan gum as supporting bath. There are many major concerns need to be addressed prior this manuscript can be accepted.

  1. The author need to state the contributions they have in this work, since the cross-linking method is common and HA-based bioprinting is widely used in many area.
  2. The author did not investigate the printing process of the embedded bioprinting the relationship between the printing parameters (like pressure and speed) and the composition of the supporting bath and bioink. Rheology experiment should be included in this research otherwise the author can not say they are doing printing research.
  3. The cell density is too low for 3D bioprinting it is hard to get any functional tissues with such low cell density. Bioprinting with much higher cell density (>3-5*106Cells/mL should be included.

Author Response

Response to Reviewer 3 Comments

Comment 1. The author need to state the contributions they have in this work, since the cross-linking method is common and HA-based bioprinting is widely used in many area.

Response 1. Thank you for the comment. As far as we know, this is the first paper describing the freeform bioprinting process involving the cascade enzymatic reaction by horseradish peroxidase and oxidase. The use of the cascade gelation process catalyzed by two enzymes enables suppressing the cytotoxicity caused by the conventional enzymatic process using horseradish peroxidase alone. HA is used as a model polymer cross-linkable through the cascade enzymatic process (Line 67-69; 87-91; 110, 111).

Comment 2. The author did not investigate the printing process of the embedded bioprinting the relationship between the printing parameters (like pressure and speed) and the composition of the supporting bath and bioink. Rheolog experiment should be included in this research otherwise the author can not say they are doing printing research.

Response 2. Thank you for the indication. In accordance with this comment and reviewer 2's comment 3, we have added data and relevant descriptions of the effects of horseradish peroxidase and choline content on the viscoelastic properties of the ink (Figure S1; Line 174-175; 298-302). As described above, the original point of the present work is that the application of the enzymatic cascade reaction of oxidase and horseradish peroxidase resulting in hydrogelation to previously reported freeform bioprinting in the support bath containing xanthan gum reported by Patrício et al in 2020 (cited in main text). It means the different point with the receding report is that the use of different hydrogelation process. Therefore, we investigated the effect of factors that affect hydrogelation on printing, and the effect of the hydrogelation process on printability and cells. The results are shown in Figures 3-6, and S2.

Comment 3. The cell density is too low for 3D bioprinting it is hard to get any functional tissues with such low cell density. Bioprinting with much higher cell density (>3-5x106cells/mL should be included.

Response 3. Thank you for the comment. In this study, we aimed to develop novel bioprinting method. To demonstrate the feasibility, we evaluated cytocompatibility of the method at low cell density. We completely agree that higher cell densities need to be applied to produce functional tissues. The cell density and the kind of cells contained in the printed constructs are strongly dependent on tissue of interest. In addition, high cell density often induces cell death due to insufficient oxygen and nutrient supply, and make it difficult to evaluate the cytocompatibility of the method itself. We believe such practical target tissues are decided as a next step based on this kind of fundamental study showing the cytocompatibility of the method. The studies at higher cell densities of human liver cells are under investigation for developing artificial liver and we will report the results in future. 

Round 2

Reviewer 2 Report

The authors have addressed all the comments and have revised the manuscript accordingly.

Reviewer 3 Report

This manuscript have improved a lot and the authors have addressed all my questions, this manuscript can be accepted for publication.